# OpenReview forum: "gRNAde: Geometric Deep Learning for 3D RNA inverse design"
_NeurIPS.cc/2024/Conference — Submitted to NeurIPS 2024_

### Official Review · Reviewer_QQ46 · 2024-07-05

**Soundness:** 4
**Presentation:** 4
**Contribution:** 3
**Rating:** 8
**Confidence:** 4

**Summary:**

gRNAde is a graph neural network designed to address the RNA reverse folding problem, a significant challenge due to the potential of RNA as therapeutic modalities and their unique data properties. RNA molecules have lower thermodynamic stability compared to proteins, resulting in fewer training samples, and their increased flexibility means multiple final states are possible. gRNAde addresses these issues by proposing a custom multi-graph representation and extending message passing to operate independently on each conformer while sharing an adjacency graph. The authors thoroughly explore evaluation techniques, comparing their model performance against Rosetta by assessing the percentage of native sequence recovery on held-out sequence families. Additionally, they demonstrate slightly improved performance when utilizing multiple conformers for model training. The authors also conduct an interesting zero-shot ranking analysis on mutation data providing a refreshing evaluation against random baselines.

**Strengths:**

- The paper is exceptionally well written presenting a thorough overview of the challenges within modality, broader field, and the importance of the problem.
- The experimental validation assesses the utility of design choices and although the improved performance in the presence of multiple conformers isn't large (the authors don't report statistical significance), the approach is promising.
- The authors conduct a variant effect evaluation assessing whether their model is capable of learning impact of single or double mutant sequences demonstrating convincing improvement over random baselines.

**Weaknesses:**

- I find the arguments regarding gRNAde perplexity being correlated with recovery to have limited support in the current presentation. In figure 2 (b) color denotes perplexity instead of one of the axis making it very challenging to assess the correlation. In addition the authors don't report a correlation value or its significance.
- The authors only use random baselines for the retrospective variant effect analysis. Including another reverse folding model or a metric from Rosetta similar to gRNAde's perplexity could strengthen the evaluation.

**Questions:**

-  Can the authors conduct a study assessing the impact of the quantity of training data on gRNAde? As structural data increases, it will be useful to understand how gRNAde's performance scales with data abundance.
- What are the other possible bottlenecks currently inhibiting further performance improvements of multi-conformational models? Do the authors believe further advancements in architectural designs are required to demonstrate larger improvements between models capturing a single vs multi-conformer information?

**Limitations:**

- The authors effectively discuss the current evaluation limitations and the difficulty of assessing the novelty and ground truth recovery of generated sequences.
- There is limited discussion on the data limitations in the current field. With only 4000 sequences, training points are very few, presenting a major challenge.
- The authors could include a brief statement on the broader impacts, such as the potential design of harmful molecules. As these models improve, the dual-use concern becomes legitimate.

---

> ### Author Rebuttal · Authors · 2024-08-06
>
> Thank you for your encouraging and actionable review! We believe our revised paper will be strengthened by incorporating your suggestions. We hope our rebuttal further addresses your questions and concerns.
>
> > Question 1
> - We have ablated the inclusion of long (primarily ribosomal) RNAs in gRNAde’s training data, which also serves to tell us how # training samples and maximum length impact performance. See Appendix D, Ablation Study and in particular the results shaded in yellow under ‘Max. train RNA length’. Appendix Figure 15(a) is also relevant to show the length distribution.
> - Number of training samples corresponding to maximum length cutoffs:
>     - cutoff @ 500 → 2607 samples
>     - cutoff @ 1000 → 2876 samples
>     - cutoff @ 2500 → 3467 samples
>     - cutoff @ 5000 → 4022 samples
> - Overall, we found that (somewhat unsurprisingly) using more data and learning from ribosomes generally improves performance. At cutoff length 500/1000, there is noticeable drop in performance. We eventually chose to report our main results for models trained on all data (cutoff at 5000) as it lead to models with the lowest perplexity, which is akin to ability to ‘fit’ the data distribution.
> - We will include these additional details from the rebuttal in the ablation study in our revised manuscript.
>
> > Question 2
> - See global response on ‘Why are performance improvements for multi-state gRNAde marginal’. In summary, the performance gains are marginal b/c of our challenging split, lack of very dynamic RNA in the training set, and difficulty of the task itself. Despite this, we believe there is some signal that multi-state architectures improve performance in both single-state and multi-state settings by explicitly extracting information about RNA dynamics.
> - We have also included results for more expressive multi-state pooling methods (Deep Symmetric Sets) in the new PDF attached – unfortunately, it did not improve performance but we believe this will be interesting to readers.
>
> > Weakness 1
> - In the new PDF attached with the rebuttal, we have added **regression plots** measuring gRNAde perplexity vs. sequence recovery as well as 3D self-consistency metrics. We made plots for both the 14 RNAs from the Rosetta benchmark in Figure 2, as well as across all 100 RNAs from the test set (16 designs each → 1600 points in each plot). We measured correlation coefficients and MAE/RMSE of regression in each plot.
> - We found weak positive/negative correlation (depending on metric) as measured by Pearson/Spearman correlation coefficients of +- 0.4 to 0.5. Recovery is more correlated than structural self-consistency (which is also harder to measure due to limitations of the structure predictor itself).
> - Besides correlation, the visualizations also suggest that perplexity < 1.2 is indicative of good designs in terms of both recovery and structural self-consistency. There are clearly far more good designs at low perplexities below 1.2 compared to higher values.
> - Thanks for highlighting this – **we think it will make for an important addition to the revised manuscript**.
>
> > Weakness 2
> - We acknowledge your point that we could have used stronger baselines than random, but our goal was to show a new capability of gRNAde that (to our knowledge) **has not been explored at all for RNA previously** in the literature.
>
> > On limitation point 2
> - We have noted that paucity of RNA structural data is a major limitation and challenge for modellers in our Introduction.
> - Obviously, more data is better, but the inverse folding task is inherently local so we should be thinking about quantity of data in terms of # of unique tokens/nucleotides (different from structure prediction, which is a more global task). In that regard, we feel 4K RNAs with an average length of 100-150 nucleotides is a sufficiently large dataset to develop at least a **useful** inverse folding tool for the community (eg. outperforms Rosetta and can be useful for fitness ranking, too).
>
> > On limitation point 3
> - That’s a fair point – we will append the societal impact section of the NeurIPS paper checklist (point #10) to make a statement on potential dual-use: “We hope that our tools contribute to the development of RNA-based therapeutics towards improving health outcomes and biotechnology applications. However, it is worth noting that generative models for biomolecule design can be misused for the development of harmful molecules for negative use cases.”

---

> > ### Comment · Reviewer_QQ46 · 2024-08-10
> >
> > Thank you to the authors for additional experiments and explanations. I think the paper is very well written, and expect that the additional resources in the form of notebooks and other code provided by the authors, can spur further innovation on this important problem.
> >
> > It's interesting that across all the author evaluations, there is a sharp performance increase at perplexity < 1.2. This is a great work and I recommend acceptance. I have increased the overall score

---

> > > ### Author Response · Authors · 2024-08-12
> > > **Thank you for responding**
> > >
> > > Thank you for responding to the rebuttal. We're happy to hear that it further addressed your concerns.

---

### Official Review · Reviewer_Hi6M · 2024-07-07

**Soundness:** 3
**Presentation:** 3
**Contribution:** 2
**Rating:** 5
**Confidence:** 3

**Summary:**

This paper proposes a geometric RNA design model. Specifically, it introduces multi-stage GNN to encode multiple conformations and aggregate these candidates, and further feed decoder to predict probabilities of a set of candidate sequences.

**Strengths:**

1.	This work creates a new dataset for RNA inverse design, with diverse properties such as sequence length, number of structures, and structural variations.
2.	This work designs a multi-state RNA reverse design model, distinct from existing methods.

**Weaknesses:**

1.	The technical contribution appears to be somewhat weak. The backbone used is from existing GNN models for equivariant design.
2.	Some technical details are not claimed. For example, are the comparison baselines retained on the new dataset or simply tested on their released model.

**Questions:**

1.	It is nice that they compared several SOTA baselines. However, I'm not sure if they 're retraining these baselines on the new dataset or just testing their released model.
2.	If I understand the claimed multi-state concept correctly, they are multiple conformations of a single RNA as input. It seems that multi-state design is an ensemble fusion of candidate representation. So, for a specific RNA with multiple conformations, does multi-state design always achieve better metrics than single-state design?
3.	I am still concerned about the technical insights of this paper as NIPS is highlight on the technical contributions. And I note that this paper has been accepted in ICML23 workshop.

**Limitations:**

This paper has no negative societal impact.

---

> ### Author Rebuttal · Authors · 2024-08-05
>
> Thank you for your actionable review. We think details in our appendix and rebuttal responses address several of your questions and concerns – please do consider revising your score if you find the responses satisfactory and let us know if there is something we can further clarify.
>
> > Weakness 1 and Question 3
> - See global response on ‘Lack of architectural novelty’.
> - The other reviewers have positively noted the following **new technical contributions**:
>     - Careful data preparation and experimental setup as well as evaluation (vioU, h3v6)
>     - New design capabilities: Improved inference speed and accessibility over physics-based tools for RNA design (h3v6); Zero-shot mutant fitness ranking (E3ka, QQ46)
> - Please reconsider the suitability of our work for NeurIPS. The official Call for Papers clearly states: “We invite submissions presenting new and original research on topics including but not limited to the following: … **Applications**, **Machine learning for sciences**.” We believe our work is well within the scope of NeurIPS.
>
> > Weakness 2 and Question 1
> - For the experiments in Figure 2, Rosetta and the others methods are not ML models but rather physics based software which **do not require training**. We reported the numbers stated in [Das et al. 2010](https://www.nature.com/articles/nmeth.1433), Supplementary Table 1. In more detail:
>     - As noted in line 236-237, we have followed the evaluation protocol established by Rhiju Das et al. in their Nature Methods paper which first pioneered 3D RNA design: (1) we have evaluated our model on the same set of 14 high quality RNA 3D structures of interest from the PDB, and (2) for the physics-based baselines, we have reported the numbers from Das et al. 2010. See Appendix E, Table 2 for full results per RNA.
>     - The ViennaRNA 2D-only baseline has been evaluated by us using the ViennaRNA python package. This is a thermodynamics-based method which does not involve any training, either. We will make a note of this in our revision – thank you!
>     - In summary, the only method requiring training is gRNAde (our work), and we have carefully prepared data splits to evaluate for generalization and ensure fair comparison to these classical baselines (detailed in line 201 onwards).
> - All the other experiments involve baselines and models that have been trained from scratch by us using the same experimental settings and evaluation protocols as our final model.
> - **Please let us know any other minor/major experimental details that you felt were missing**. We will promptly clarify – we want to make the experimental protocol and setup completely transparent and reproducible (along with a detailed and documented codebase).
>
> > Question 2
> - Your understanding of the multi-state setting is exactly correct!
> - Yes, we show that multi-state gRNAde architectures marginally improve performance for multi-state RNAs in Figure 4:
>     - Figure 4(a) shows improvements in aggregated performance across 100 test set samples (all of which have multiple conformational states).
>     - Figure 4(b) shows **why** there is improvement: at a per-nucleotide level, multi-state gRNAde has better performance for nucleotides that are locally more flexible, undergo changes in base pairing within multiple states, and are on the surface.
> - We would also like to share another interesting finding: multi-state gRNAde also improves performance on the single-state test set. See Appendix D, ‘Ablation Study’, line 600 onwards (in the table, see the columns for sequence recovery as well as 2D/3D self-consistency for the ablated variants highlighted in red). Thus, even for test set RNAs that have only one state available, training gRNAde in a multi-state manner can lead to better inverse folding performance.
> - What do we understand/take away from these results? Multi-state training seems to allow gRNAde to better understand RNA dynamics and conformational changes.

---

> > ### Comment · Reviewer_Hi6M · 2024-08-12
> >
> > My main concerns are addressed. Thanks for your responses.

---

> > > ### Author Response · Authors · 2024-08-12
> > > **Thank you for responding**
> > >
> > > Thank you for acknowledging the rebuttal -- we are happy to hear your main concerns are now addressed.
> > >
> > > Would you consider increasing your score to reflect that?

---

> > > > ### Author Response · Authors · 2024-08-14
> > > > **Are all your concerns addressed?**
> > > >
> > > > Dear Reviewer Hi6M,
> > > >
> > > > As you stated that your concerns are addressed, would you consider revising your score to reflect that?
> > > >
> > > > Best regards,
> > > >
> > > > Authors

---

### Official Review · Reviewer_E2ka · 2024-07-11

**Soundness:** 2
**Presentation:** 3
**Contribution:** 2
**Rating:** 4
**Confidence:** 4

**Summary:**

This paper introduce gRNAde, a geometric deep learning pipeline for RNA sequence design conditioned on one or more 3D backbone structures. gRNAde is superior to the physically based Rosetta for 3D 320 RNA inverse folding in terms of performance, inference speed, and ease of use. The method demonstrates significant superiority across various experiments.

**Strengths:**

1. The authors introduce gRNAde, the first work to consider multi-state biomolecule representation. This study explores the feasibility and specific experimental results of using multi-state biomolecule representation, providing new ideas for researchers in the field.
2. The authors conduct extensive experiments and analyses on multiple datasets and experimental settings, demonstrating the model's effectiveness from various perspectives, especially regarding the "Zero-shot ranking of RNA fitness landscape" experiment, which is currently lacking in this field.
3. The authors present various experimental details using numerous visualizations and data tables, making the paper easier for readers to understand.

**Weaknesses:**

1. The model architecture proposed by the author lacks innovation. The core structure of gRNAde is directly stacked using GVP-GNN, and the handling of multi-state conformations is merely simple stacking. Additionally, the 3-beads representation method is very common in traditional RNA 3D structure modeling, which is also not an innovation by the author. Therefore, I believe the model design is lacking.
2. The baselines compared by the author in various experiments are either outdated or too simple, such as "Rosetta(2020)" and the "random baseline" in the Zero-shot experiment. This makes it difficult to demonstrate the actual performance of gRNAde. Some recent works using deep learning to model RNA 3D structures can serve as baselines, such as [1-3].
3. The author mentions that gRNAde has a significant speed improvement over Rosetta, but the author did not run the Rosetta code themselves and instead directly cited the original Rosetta paper. I believe this point is debatable because the model's running speed is also limited by GPU computational performance. The author uses an A100, whereas the GPU used by Rosetta four years ago is obviously inferior to the A100. Therefore, the author needs to rerun the Rosetta program on the A100 to provide accurate model inference times.







[1] Geometric deep learning of RNA structure, Science 2021

[2] Physics-aware Graph Neural Network for Accurate RNA 3D Structure Prediction, NIPS workshop 2022

[3] RDesign: Hierarchical Data-efficient Representation Learning for Tertiary Structure-based RNA Design, ICLR 2024

**Questions:**

1. The second experiment results indicate that multi-state biomolecule representation did not significantly improve performance. What does the author believe is the reason for this?
2. The RNA used in the first experiment are older and limited in number. Has the author considered testing with more recently published and more numerous RNA?

**Limitations:**

The authors discuss practical tradeoffs to using gRNAde in real-world RNA design scenarios 330 in Appendix B, including limitations due to the current state of 3D RNA structure prediction tools.

---

> ### Author Rebuttal · Authors · 2024-08-06
>
> Thank you for your review – please see our detailed responses below – we believe we have addressed several of your concerns and questions. Please let us know what further information we can provide to make you reconsider your vote to reject the paper.
>
> > Question 1
> - See global response ‘On marginal improvements of multi-state gRNAde’. In summary, the performance gains are marginal b/c of our challenging split, lack of very dynamic RNA in the training set, and difficulty of the task itself. Despite this, we believe there is some signal that multi-state architectures improve performance in both single-state and multi-state settings by explicitly extracting information about RNA dynamics.
>
> > Question 2
> - The 14 RNAs that are used to compare to Rosetta in Figure 2 are indeed a decade old, but we believe they are still a great benchmark because:
>     - They are extremely high resolution crystal structures (even by today’s standards for RNA).
>     - They were handpicked by Rhiju Das in their [Nature Methods paper](https://www.nature.com/articles/nmeth.1433) as being RNAs which are interesting for biologists to design and develop new versions of.
> - In addition to the above, in Appendix D, we report a more comprehensive set of results on our full test set of 100 RNAs in total (Single-state split). This includes the 14 from Figure 2 as well as all recently released versions of the same RNAs and their structural homologues (see line 201 onwards for precise description of the splits).
> - In summary, the model’s performance does not degrade when tested on more recent and greater quantities of RNA.
>
> > Weakness 1
> - See global response on ‘Lack of architectural novelty’. We have also included results for more expressive multi-state pooling methods (Deep Symmetric Sets) in the new PDF attached – unfortunately, it did not improve performance but we believe this will be interesting to readers.
>
> > Weakness 2
> - Some comments on your suggestions:
>    - *Geometric deep learning of RNA structure* – this is a structure ranking model that takes as input an RNA structure + RNA sequence, and outputs a score for how closely that structure resembles the true structure (it actually predicts RMSD) – such structure prediction and ranking models cannot be used for inverse design.
>     - *Physics-aware Graph Neural Network for Accurate RNA 3D Structure Prediction* – similarly, this is also a model for structure prediction/ranking and it is inapplicable to the inverse folding problem. We will cite it in our revision.
>     - *RDesign: Hierarchical Data-efficient Representation Learning for Tertiary Structure-based RNA Design* – please see global response on ‘Comparison to RDesign’ for why apples-to-apples comparison to their work is not possible.
> - We have done a literature review in Appendix A where we have discussed deep learning for RNA structure modeling, why current tools cannot be used for inverse design, and how gRNAde is contextualized within the broader literature.
> - We believe Rosetta is a highly relevant baseline as it is the state-of-the-art in physics-based modeling of RNA. Yes, we realize that the Rosetta results are a decade old, but the reason for this is that 3D RNA design has not received as much attention from the Rosetta community as 3D protein design.
>
> > Weakness 3
> - Unfortunately, it is not possible to use RNA design recipes in the latest Rosetta builds (we did try).
> - A major limitation of Rosetta recipes is that many of them do not use GPUs (and this is a major advantage of new deep learning based alternatives). Rosetta RNA design recipes do not use GPUs, which is why they are so slow. Most of the Rosetta recipes are also just inherently slow b/c they use MCMC sampling and need to iterate until convergence, whereas deep learning models are one-shot predictors/generators.
> - [Tmol](https://github.com/uw-ipd/tmol) is an ongoing effort by Institute of Protein Design to port Rosetta functionality to GPUs in a differentiable manner. However, as you can see from the github this is a very early effort which is still in development without any documentation.
> - **This is all the more reason for releasing gRNAde in an open source manner and easy to access via notebooks and tutorials.** We hope that making these datasets and tools more broadly accessible will invite renewed attention to 3D RNA design.

---

> > ### Comment · Reviewer_E2ka · 2024-08-12
> >
> > Thank you for your detailed response. However, I still have the following concerns:
> >
> > 1. In fact, a comparison with the three papers I mentioned is feasible, even though the tasks they address differ from yours. You use a stacked GVP architecture, but theoretically, the GVP could be replaced with the architectures from those three papers. The fundamental difference lies in the models used to represent the RNA 3D structure. As for RDesign, the lack of training code and the specifics of the data split do not prevent the transfer of the model architecture. Therefore, I believe that simply comparing with a random baseline is not persuasive.
> >
> > 2. Regarding the use of DSS, it essentially applies an existing method (DSS) to your dataset, but this does not constitute an innovation in your model architecture. In my view, this paper still lacks sufficient innovation in terms of the modeling approach.
> >
> > 3. Concerning the use of Rosetta, while it is true that most methods within Rosetta rely on MCMC sampling and run on CPUs, this means that you cannot claim faster performance in comparison, as that would be unfair. You should compare your methods with other deep learning methods which run on GPU.
> >
> > 4. For the 3D RNA inverse design task, you are not the first to define or provide a dataset for this problem. RDesign also addresses this task and presents a more novel modeling approach. The main difference lies in the use of the multi-state (please correct me if I misunderstood).
> >
> > If I have misunderstood any of the above points, I would greatly appreciate it if you could clarify. If you can provide more detailed answers to these questions, I would be happy to consider raising my score.

---

> > > ### Author Response · Authors · 2024-08-12
> > > **Thank you for responding**
> > >
> > > Thank you for acknowledging the rebuttal and wanting to engage in discussion.
> > >
> > > > On point 1
> > > - We have in fact ablated the architecture of the GVP GNN used within our model. Please see Appendix D, line 587 onwards and the results in green. We compared using a rotation invariant GNN vs. rotation equivariant GNN, which we believe is the **fundamental concept when building geometric 3D GNNs** that should bring insight to readers (as opposed to what specific choice of layers one makes which differs a lot from paper to paper).
> > > - On RDesign, we disagree -- we have seen repeatedly in structural biology that **using random data splits will give the impression that models are working well, but that their o.o.d. performance will be greatly over-exagerated** (examples: [1](https://arxiv.org/abs/2308.05777) [2](https://openreview.net/forum?id=A8pqQipwkt) [3](https://arxiv.org/abs/2206.12411) -- it is very simple to delude ourselves with random splits).
> > >     - RDesign seems overfit on its training data based on [the logs](https://github.com/A4Bio/RDesign/blob/master/checkpoints/log.log): their training perplexity is order of magnitude lower than validation perplexity even on random splits...
> > >     - Running RDesign on our splits is guaranteed to have data leakage.
> > > - Also, **we did ablate the fundamental conceptual difference between RDesign and gRNAde** -- autoregressive vs. non-autoregressive decoding -- in Appendix D, line 591 onwards. We find that autoregressive decoding is better for structural self-consistency, which we care more about in real-world design scenarios. We think this finding will be super interesting to the community working on other types of biomolecules, too, as their is debate on the two approaches to decoding.
> > > - Finally, if you really want to compare numbers, our model's recovery rates are in the 50s for the single-state split. RDesign's recovery rates are in the 40s in their Table 2.
> > >
> > > > On point 2
> > > - Adapting existing architectures to new problems, doing the experiments rigorously, setting the first benchmarks for a field, and releasing open-source resources so others can build upon them is a valuable contribution. This is stated in the NeurIPS Call for Papers, too.
> > > - It is valuable for the community to know how to apply our best tools work to new problems. Doing applied work well requires strong understanding of the application domain as well as the deep learning architectures and evaluation, but may not always involve new architecture ideas.
> > > - It would obviously have been nice for the novelty of our paper if we could have proposed a fancy multi-state fusion method. However, we benchmarked many ideas rigorously and did not find them to bring improvements over Deep Set. We should not be penalized for this.
> > >
> > > > On point 3
> > > - The point of those experiments is to demonstrate that deep learning tools perform better and are **more broadly accessible** that Rosetta RNA (its latest build cannot even run RNA design...). We tried to compare as fairly as possible to Rosetta by being careful about the splits.
> > > - There is a lot of precedent for deep learning papers claiming superior speed and performance to Rosetta in the same manner as us for other tasks, such as inverse folding, structure generation, rotamer optimization, etc. (Examples [1](https://www.science.org/doi/10.1126/science.add2187), [2](https://www.mit.edu/~vgarg/GenerativeModelsForProteinDesign.pdf), [3](https://www.pnas.org/doi/full/10.1073/pnas.2216438120) and many structure prediction papers rely on this claim)
> > > - We do take your point though and we will be more careful about our Rosetta speed claim. We propose to revise it to add caveats everywhere in our paper, for instance:
> > >     - Line 60: "gRNAde is significantly faster than Rosetta for inference; e.g. sampling 100+ designs in 1 second for an RNA of 60 nucleotides on an A100 GPU, compared to the reported hours for Rosetta **on CPUs, making gRNAde more broadly accessible**.
> > >     - Line 246: "Rosetta takes order of hours to produce a single design due to performing expensive Monte Carlo optimisations **on a CPU**."
> > >
> > > > On point 4
> > > - RDesign and our work were developed concurrently but **they beat us to publication**. These kinds of situations are very difficult for authors and we have tried to provide an honest and detailed comparison between the two works in line 507 onwards as well as the ablation study experiments we already pointed you to.
> > > - RDesign did not release training code -- this makes it impossible to reproduce their study.
> > > - RDesign's evaluation makes use of random splits (we already highlighted why this is has issues), does not compute any metrics beyond recovery, and does not compare to Rosetta or physics-based tools.
> > > - We think our study brings significant new insights to the community and can hopefully be complementary to the RDesign paper. We believe these are concurrent works and one being published first should not be the reason to stop the publication of the other.

---

> ### Author Response · Authors · 2024-08-12
> **Now trying to run RDesign**
>
> We also wanted to add an additional note: After your comment, we are now trying to run RDesign to see if we can load the datasets and checkpoint, as well as run inference with it.
>
> 1. There are **no installation instructions** available in the repository: https://github.com/A4Bio/RDesign/
>
> 2. The dataset files released by them are **corrupted** (we tried loading them on a Macbook, a linux server, and on Github Codespaces). You can even try it quickly on your end on your browser:
>     - Open a GitHub Codespaces on their repository.
>     - Download the dataset files they have released: https://github.com/A4Bio/RDesign/releases/tag/data
>     - Upload any of the files to the Codespaces.
>     - Open a terminal or create a new jupyter notebook in the Codespaces and type the following:
> ```
> import torch
> torch.load("<path-to-data>/<data_file>.pt")  # or torch.load("<path-to-data>/<data_file>.pt", map_location='cpu')
> ```
> ...which will always lead to the following error:
> ```
> RuntimeError                              Traceback (most recent call last)
> Cell In[3], line 1
> ----> 1 torch.load("val_data.pt", map_location='cpu')
>
> File ~/.local/lib/python3.10/site-packages/torch/serialization.py:1040, in load(f, map_location, pickle_module, weights_only, mmap, **pickle_load_args)
>    1038     except RuntimeError as e:
>    1039         raise pickle.UnpicklingError(UNSAFE_MESSAGE + str(e)) from None
> -> 1040 return _legacy_load(opened_file, map_location, pickle_module, **pickle_load_args)
>
> File ~/.local/lib/python3.10/site-packages/torch/serialization.py:1264, in _legacy_load(f, map_location, pickle_module, **pickle_load_args)
>    1262 magic_number = pickle_module.load(f, **pickle_load_args)
>    1263 if magic_number != MAGIC_NUMBER:
> -> 1264     raise RuntimeError("Invalid magic number; corrupt file?")
>    1265 protocol_version = pickle_module.load(f, **pickle_load_args)
>    1266 if protocol_version != PROTOCOL_VERSION:
>
> RuntimeError: Invalid magic number; corrupt file?
> ```
>
> This happens for every data file they have provided.
>
> We will keep you posted if we manage to run the code, but these two points should already tell you why direct apples-to-apples comparisons are so difficult in this situation. Without being able to load the processed dataset files, and without any documentation in the README or within the code as to what is the expected format of input data to their model, it is not possible to re-train or do inference with this model.

---

> ### Author Response · Authors · 2024-08-12
> **Further issues with RDesign**
>
> Another issue that makes direct comparison with RDesign impossible is that **their model does not implement sampling** during inference.
>
> - Their model class contains a method called `sample()`: https://github.com/A4Bio/RDesign/blob/master/model/rdesign_model.py#L76 -- however, this method does not actually implement any sampling from the probability distribution outputted by the model. It just directly outputs the probability distribution.
>
> - This would explain why RDesign's perplexity numbers are non-sensical (Appendix E.3 of their paper, as well as just the training logs for their model).
>
> - Not being able to perform sampling means that the model will be useless in real design scenarios (especially as it is non-autoregressive/one-shot independent decoding per token), because it will keep outputting the same probability distribution and inherently cannot be used to generate diverse sequences as a result.
>
> Is the reviewer somewhat convinced now that:
> 1. Our work brings something new to the community and is a valuable contribution?
> 2. Direct comparison to RDesign is not possible given the limitations of both their experimental methodology as well as their reproducibility?

---

> > ### Author Response · Authors · 2024-08-13
> > **Got RDesign to work; results are very poor**
> >
> > After a lot of effort, we figured out how to make the RDesign code perform inference.
> >
> > We have made direct comparisons to RDesign in this comment: https://openreview.net/forum?id=Fm4FkfGTLu&noteId=RdkCCdrrqx - RDesign underperforms gRNAde and Rosetta.

---

> > > ### Comment · Reviewer_E2ka · 2024-08-14
> > >
> > > Thank you for your response!
> > >
> > > Firstly, regarding the first point, when modeling 3D structural data, equivariant neural networks generally outperform invariant neural networks, which aligns with the experimental results for the multi-state scenario in your table. However, this is not the case for the single-state scenario, and this discrepancy might require a more detailed explanation from you.
> > >
> > > Regarding the RDesign paper, the authors established strict dataset partitions to prevent potential data leakage. I would like you to retrain their model architecture on your dataset, rather than directly testing using their checkpoint.
> > >
> > > On the second point: Since RDesign was published in ICLR 2024, there has been a considerable time gap before the NeurIPS 2024 submission deadline. Therefore, your study is not the first to address the RNA inverse folding problem. I do agree with the insights and related experiments you've conducted utilizing pre-existing model architecture on the multi-state problem, but I believe this alone may not be sufficient to support a top-tier conference paper.
> > >
> > > For the third point, thank you for your response. I believe my concerns have been adequately addressed.
> > >
> > > Finally, on the fourth point, I would like to revisit RDesign. This method first categorizes RNA structures based on their similarity and then partitions the dataset based on these categories. In my opinion, evaluating RDesign’s performance reasonably requires retraining their model architecture on your dataset. Directly using the checkpoint files is not appropriate.

---

### Official Review · Reviewer_h3v6 · 2024-07-12

**Soundness:** 3
**Presentation:** 4
**Contribution:** 2
**Rating:** 5
**Confidence:** 3

**Summary:**

This work designed gRNAde, a geometric deep learning pipeline for RNA sequence design conditioned on one or more 3D backbone structures. To achieve this, the authors created single-state and multi-state 3D RNA structure datasets, built a geometric graph representation, and proposed an architecture consisting of a multi-state GNN encoder, a pooling layer, and a autoregressive decoder. The single-state RNA design, multi-state RNA design, and zero-shot rank experiments were conducted and results show that gRNAde outperformed all previous methods including Rosetta.

**Strengths:**

1.	The datasets were carefully designed. Only structures with high resolution were maintained. Two kinds of clusters were used to split train, validate, and test sets where the hard samples were split into test sets.

2.	The model architecture makes use of information from multiple conformations. This is achieved by sum or average pooling.

3.	The experiments were conducted fairly. Datasets were split carefully. The results were averaged on 16 sampled sequences across 3 random seeds.

4.	The inference speed is much faster than traditional methods. This makes it possible to be used in High throughput screening. The zero-shot ranking ability is also an advantage.

**Weaknesses:**

1.	The model architecture has no novelty. All components are token from previous work and the overall structure is very similar to that of ProteinMPNN. The multiple conformations are processed independently and the representations are simply averaged or summed, which may not grasp all information.

2.	The model is trained on only about 4 thousand RNA sequences. These sequences are too few to cover the entire space. RNAs with no 3D structures should be exploited, as done in alphafold3.

3.	The results about single-state RNA design were reported on only 14 samples. More samples should be used to test the model. For example, the results on test sets with 100 samples should be reported.

4.	The improvements on multi-state RNA design task are limited. The native sequence recovery is obviously lower than the results of single-state design task.

**Questions:**

1.	Lots of the RNA conformations come from protein-RNA complexes and DNA-RNA hybrids, which means these conformations do not appear alone. Since the model encoder cannot consider other molecules, the extracted representations are biased. How about adding a scalar feature to indicate if a nucleotide is on the interface with another molecule?

2.	Why is the decoder only designed for autoregressive decoding? The arbitrary decoding in ProteinMPNN is useful when part of the sequence is already known.

**Limitations:**

The representation ability of gRNAde remains to be verified. Usually, the representation is extracted when all nucleotides are known, so the architecture for inverse folding problem may not suitable for representation learning.

---

> ### Author Rebuttal · Authors · 2024-08-06
>
> Thank you for your actionable review. We hope that our rebuttal answers your questions sufficiently and makes you reconsider some points that you noted as weaknesses. Please do consider revising your score if you find the responses satisfactory and let us know if there is something we can further clarify.
>
> We would also like to incorporate your suggestion on arbitrary decoding order into our revision -- thank you for it!
>
> > Question 1
> - We can certainly do that and we imagine this will improve models’ performance at designing interfaces. We are developing the next version of gRNAde which goes beyond just scalar features and instead considers nodes from the interaction partners together with the RNAs nodes. We hope to use this model for binding-related tasks such as aptamer design. However, this is still **work in progress**.
>
> > Question 2
> - It is very simple to do arbitrary decoding orders for autoregressive decoding b/c we can simply permute the inputs during training and inference. And we agree that arbitrary decoding order is extremely useful in real design scenarios where we are usually given partial sequences.
> - In the new PDF attached with the rebuttal, we have added results for gRNAde trained with random decoding order (shaded in green) and seen that this leads to only minor decreases in sequence recovery and 3D self-consistency. Perplexity gets significantly worse as it is more challenging for the model to fit the training data. The same observations were made in the ProteinMPNN paper.
> - Thanks for the suggestion – **we will include these results in our updated manuscript.**
>
> > Weakness 1
> - See global response ‘On architectural novelty’. We have also included results for more expressive multi-state pooling methods (Deep Symmetric Sets) in the new PDF attached – unfortunately, it did not improve performance but we believe this will also be interesting to readers and want to include it in the ablation study.
>
> > Weakness 2
> - Obviously, more data is better, but the inverse folding task is inherently local so we should be thinking about quantity of data in terms of **# of unique tokens/nucleotides** (different from structure prediction, which is a more global task). In that regard, we feel 4K RNAs with an average length of 100-150 nucleotides is a sufficiently large dataset to develop at least a **useful inverse folding tool for the community**. As evidence of this:
>     - gRNAde performs better than Rosetta, the best physics based tool for 3D RNA design.
>     - gRNAde was found useful for ranking Ribozyme fitness in our retrospective study.
>     - Based on the Ribozyme retrospective study, we felt confident enough to send gRNAde’s designed sequences for the same Ribozyme for experimental validation to our collaborators.
> - AlphaFold 3 only came out 1 week prior to the NeurIPS deadline – we will certainly consider using their ideas in our future work but please don’t hold it against us for not using RNAs without 3D structure!
>
> > Weakness 3
> - We have already done what you have asked. We elaborate below:
> - Figure 2 does use only 14 RNAs to compare to Rosetta, but we believe they are still a great benchmark because:
>     - They are extremely high resolution crystal structures (even by today’s standards for RNA).
>     - They were handpicked by Rhiju Das in their [Nature Methods paper](https://www.nature.com/articles/nmeth.1433) as being RNAs which are interesting for biologists to design and develop new versions of.
> - In addition to the above, in Appendix D, we report a more comprehensive set of results on our full test set of 100 RNAs in total (Single-state split). This includes the 14 from Figure 2 as well as all recently released versions of the same RNAs and their structural homologues (see line 201 onwards for precise description of the splits).
>
> > Weakness 4
> - See global response ‘On marginal improvements for multi-state gRNAde’. In summary, the performance gains are marginal b/c of our challenging split, lack of very dynamic RNA in the training set, and difficulty of the task itself. Despite this, we believe there is some signal that multi-state architectures improve performance in both single-state and multi-state settings by explicitly extracting information about RNA dynamics.
>
> > On the stated limitation
> - We agree that inverse folding generative models are not suitable for representation learning/predictive tasks. It has also been seen for proteins that models for inverse folding are not useful for representation learning (eg. [this paper](https://www.biorxiv.org/content/biorxiv/early/2022/11/21/2022.05.25.493516.full.pdf) by Kevin Yang) -- our goal is inverse design and not representation learning.

---

> > ### Comment · Reviewer_h3v6 · 2024-08-12
> >
> > Thanks a lot for taking the time and effort to answer my questions, but I want to keep my neutral score. Although the performance seems ok, I still think the contribution of this paper is more like an extension of existing architecture on RNA-related tasks.This neutral score means that if other reviewers and AC lean to accept this work, I would not oppose it.

---

> ### Author Response · Authors · 2024-08-12
> **Thank you for responding**
>
> Thank you for acknowledging the rebuttal. We hope you will reconsider, given that we put significant effort to address your review and we think it did improve the paper overall.
>
> Adapting existing architectures to new problems, doing the experiments rigorously, setting the first benchmarks for a field, and releasing open-source resources so others can build upon them **is a valuable contribution**. The NeurIPS Call for Papers states: “We invite submissions presenting new and original research on topics including but not limited to the following: … **Applications**, **Machine learning for sciences**.” Doing applied work well requires strong understanding of the application domain as well as the deep learning architectures and evaluation, but may not *always* involve new architecture ideas.
>
> We addressed as many of your weaknesses and questions as we could via the rebuttal.
> - Weakness 1: We tried more expressive set pooling methods, but it often happens in structural biology applications that complex architectural ideas **do not generalize** to o.o.d. test sets. It would obviously have been nice for the novelty of our paper if we could have proposed an exciting, new multi-state fusion method. However, we benchmarked many ideas rigorously and did not find them to bring real improvements.
> - Weakness 2: We basically processed and prepared ML-ready datasets from **all the structured RNAs available publicly** on the internet; we couldn't use the AlphaFold3 self-distillation idea b/c it was not available when we did the work. We hope this is not counted as a weakness in your assessment.
> - Weakness 3: We actually did use a lot more test samples and reported results with confidence intervals in the appendix. We hope this is also not counted as a weakness in your assessment.
> - We can use our models with arbitrary decoding order, too. We did this experiment that you asked for, too.

---

### Official Review · Reviewer_vioU · 2024-07-13

**Soundness:** 2
**Presentation:** 3
**Contribution:** 3
**Rating:** 6
**Confidence:** 4

**Summary:**

This paper introduced a multi-state geometric graph neural network for the RNA inverse folding problems. Experiments are conducted on carefully splited structural datasets that avoid data leakage. The results have shown convincing performance improvement over the physics-based methods such has FARFAR and Rosetta which are commonly used for RNAs.

**Strengths:**

- This paper is well written and pleasing to read. Explanations on key biological concepts related to RNAs, and how they motivate the model design as well as experiment setup are adequate and above all, clear.
- Evaluation metrics are well considered. The self-consistent scores on the secondary and tertiary levels are meaningful.
    - limitations on self-consistence scores are acknowledged in the main text. For RNAs, many challenges are unique especially when they are compared to proteins. Therefore, clarifications and precautions are particularly needed for RNA related tasks. I appreciate the authors’ effort for stating these limitation clearly in the main text.

**Weaknesses:**

- Comparison to contemporary deep learning models for RNA inverse folding is limiting
    - RDesign (https://openreview.net/forum?id=RemfXx7ebP) for example is a recent deep learning based method for 3D RNA inverse design
    - For inverse design on the secondary structure level there are many more options — a lot of them are better than RNAinverse from ViennaRNA. I would suggest checking out this survey (Design of RNAs: comparing programs for inverse RNA folding) and include a few other more competitive baselines.
- For the self-consistency scores, I personally doubt if RhoFold (also called e2efold-3d) is reliable software for RNA tertiary structure prediction, since it is from the same group that published e2efold which is a spectacularly awful RNA secondary structure predictor (I personally would avoid using any of their tools; checkout its Github issues, and also followup works on RNA secondary structure predictions that have compared with e2efold). Have the authors used more recent folding softwares such as RosettaFoldNA and AlphaFold3?
    - Would using different structure predictors significantly impact the results? This also includes EternaFold. How would gRNAde hold up against the baselines when RNAfold or LinearFold is used to compute the self-consistency scores on the secondary level?
- Data splits (train, validation and test) are carefully constructed so that the evaluation is not contaminated by data leak. But I wonder if the TM-score cut-off at 0.45 is too lenient? Is it still possible to have similar structures between training and test sets under this threshold?

Line 258 to 259. The argument would be more compelling if the inverse design operates at the quaternary level which would include information about ligand structures.

**Questions:**

- Why would a perfect model has a perplexity of 1 (i.e. mapping each structure to a single sequence)? The solution for RNA inverse folding should be an one to many mapping, because for a target backbone structure there are potentially many sequences that can fold into the same structure. Likewise, a perplexity of 4 doesn’t mean the predictions are nonsense. For unfolded RNAs (e.g. a linear chain), they very likely correspond to random RNA sequences.
- Conformers would indicate that the structures are the local energy minimizers on the potential energy surface. How would you know if these multi-state conformations are really conformers? Where are multi-state conformations from?

**Limitations:**

- Comparison to contemporary models for RNA inverse folding is a bit hollow. It would be more meaningful if some deep learning based baselines can be included into the comparison.

---

> ### Author Rebuttal · Authors · 2024-08-05
>
> Thank you for your encouraging review! We think that incorporating your comments will strengthen our revised paper and we hope our responses address your questions in the best way possible.
>
> > Question 1:
> - To clarify, ‘perfect’ in this sentence is from a machine learning context, not from an applied context. So given an RNA backbone as input, a perfect ML model will output its groundtruth sequence from the PDB, which it would have memorized (leading to perplexity = 1).
> - However, such a perfect ML model is completely useless from an applied perspective, because we never want to perfectly recover the groundtruth sequence from the PDB. We want to sample a diverse set of designs which are reasonably close to the PDB sequence while remaining interesting/non-trivial.
> - Perplexity = 4 would mean that the model is randomly selecting from the 4 bases for each position in the sequence, so your understanding is correct.
>
> > Question 2:
> - We re-visited basic chemistry definitions and agree that ‘conformer’ is a term reserved for conformational states at which a molecule's energy is at a local minimum. **We will revise our paper to not use this term** – thank you!
> - The multiple conformations that we use are from multiple deposited structures for a given PDB entry, or multiple PDB entries for the same RNA, or both. The answer to whether they are at local energy minima really depends on who one asks, in our opinion. Our current understanding is that it remains an open question whether crystal structures of biomolecules deposited in the PDB are truly energy minimas or just some frozen states/artifact of crystallization.
> - See Appendix Figure 15 for some statistics on # of multi-state RNAs, # of states per RNA, whether # of states is correlated with length (no), etc.
>
> > Weakness 1 and Limitation 1:
> - Please see the global response on ‘Comparison to RDesign’ for why apples-to-apples comparison is not possible. We have also done extensive ablation studies of our architecture as a way to understand the impact of different components on performance.
> - 2D-only design techniques are meant for different types of RNA design problems than what Rosetta/gRNAde are intended for. They can not incorporate 3D information about kinks and turns and motifs explicitly (b/c they optimize for only maintaining the same base pairing in designed sequences; they are also not evaluated for ‘recovery’ but rather for a ‘success rate’ of retaining the same exact base pairing).
>     - You can see this by comparing gRNAde variants to ViennaRNA in Appendix Table 1: ViennaRNA actually has **very good** 2D self-consistency score but poor 3D self-consistency as it is simply not designed to account for 3D structure. All other 2D inverse folding methods we are aware of also cannot account for 3D interactions, so we think it is reasonable to expect similar performance.
>
> > Weakness 2:
> - AlphaFold3’s paper was published one week before the NeurIPS deadline and its code or weights are not yet available. Our framework is flexible to swap out RhoFold for AF3 as soon as possible (we are also eager to move on as there are limitations to RhoFold's performance; see the 'Groundtruth sequence prediction baseline' in our ablation study for an upper limit on our test set).
> - RF2NA is not a suitable choice as it is primarily for protein-NA complexes, not solo RNAs. We also think (intuitively) that RhoFold’s use of a language model makes it less reliant on MSAs than RF2NA (we almost never have MSAs for designed/synthetic RNAs).
> - RhoFold was used in AIchemy_RNA which was used as a baseline in AlphaFold3 as ‘the top performing machine learning system’ on CASP15 targets. See Extended Data Fig. 5 in AF3 paper, where AF3 outperforms RhoFold but not outright (RhoFold is better on some targets). **So we think RhoFold is a reasonable choice at present.**
> - We chose EternaFold b/c it has actually been evaluated on designed/synthetic RNAs in the wet lab before and is broadly accessible for benchmarking. We are aware that it is not the best 2D structure predictor for naturally occurring RNAs, but believe that it does a good job at recovering at least the correct base pairings (which is the goal of 2D self-consistency evaluation).
> - Nobody has conducted a self-consistency score based benchmark for 3D RNA design before us, so we are unable to report self-consistency numbers for Rosetta/other baselines (it is not possible to use the RNA backbone design recipe in the latest Rosetta builds).
>
> > Weakness 3:
> - Our choice of a TM-score cut-off of 0.45 to determine whether RNAs are in the same global fold follows [US-align](https://zhanggroup.org/US-align/) and other similar works on RNA structure modeling (eg. see [recent CASP evaluation for RNA structure prediction](https://onlinelibrary.wiley.com/doi/full/10.1002/prot.26602) where the 0.45 cutoff was used). Whereas it is conventional in the proteins literature to cluster protein structures according to a 0.5 TM-score cut-off, given the wider and more flexible nature of RNA compared to protein structures, a slightly lower structural similarity cut-off for RNA is warranted.
>
> > Weakness 4 / lines 258 and 259:
> - For riboswitches, the aptamer domain that is interacting with the ligand partner is usually conserved, and we currently want to use gRNAde to re-design scaffolds around this binding domain that may be more thermally stable while retaining the switching mechanism. We think gRNAde can be somewhat useful for this already.
> - But we agree that ligand conditioning during sequence design will provide greater context for multi-state design, and **we are actively developing a version of gRNAde which incorporates ligand partners**.

---

> > ### Comment · Reviewer_vioU · 2024-08-12
> >
> > I have read and appreciate the author's response.
> >
> > While I appreciate the paper's writing and its clear biological motivation and analysis, I had hoped to see a more comprehensive comparison with competitive baselines. This seems to be a shared concern among other reviewers as well.
> > - According to [the conference policy](https://neurips.cc/Conferences/2024/PaperInformation/NeurIPS-FAQ), "Papers appearing less than two months before the submission deadline are generally considered concurrent to NeurIPS submissions.", so I am not sure if [RDesign](https://openreview.net/forum?id=RemfXx7ebP) satisfies that criterion, since it was published in Jan 16 2024 (although it was modified later in Apr).
> > - But anyways, I understand that RDesign lacks sufficient utility or means of reproduction. I appreciate any efforts the authors have put into running RDesign.
> >
> > I was also hoping to see some sanity checks, in particular regarding the "consistency" of the 2D/3D self-consistency tests. Although these provide important insights into the model's performance, they are essentially based on imperfect computational models, so it is expected there will be variance in these self-consistency scores.
> > - This is the reason why, in my opnion, using a set of relatively good and reliable computational predictor for these self-consistency would provide more reliable performance evaluation.
> > - It is true that AlphaFold 3 is not publicly released but they have an online server where you can submit jobs. If they accept batch jobs, and if time permits, then I would suggest giving it a try...
> > - Now for the 2D self-consistency scores, I think it really doesn't hurt for you to give RNAfold or LinearFold a try. I doubt it would generate any significant impact on the results paper, but having these additional checks would greatly strength this paper.

---

> ### Author Response · Authors · 2024-08-12
> **Thank you for responding**
>
> Thank you for responding.
>
> Re. RDesign
> - As you noted, 2 other reviewers were concerned with the comparison to RDesign. Short of all options, we have been trying to use RDesign but we simply cannot reproduce its data format or run inference with their model, at present (eg. no installation instructions, dataset files seem corrupted, model does not have an actual sampling mode). **We hope you would agree that the community needs good open source training code, datasets, and reproducibility in order to further develop this new research direction (RNA design with deep learning).**
> - Also, please see our ablation study for apples-to-apples comparisons to RDesign's architectural ideas vs. gRNAde's (majorly, non-autoregressive/one-shot vs autoregressive), benchmarked fairly on our split and experimental settings. We think this brings insights to readers, b/c the question of autoregressive vs. one-shot models is also relevant for other biomolecule design tasks.
>
> Re. publication date:
> - The decision notifications for ICLR were released in January privately to authors. We would imagine that the actual publication date is when a camera-ready version of a paper is **released to the public audience** beyond Authors, Reviewers, ACs, PCs (which was April).
>
> Re. consistency of self-consistency metrics: That's a great point and we do agree that these metrics are not perfect. These metrics (as with most metrics for generative models) are supposed to be proxies for actually evaluating these designs in real world experiments. So maybe they are not perfect, but the way we actually use them for RNA design is as filters where really poor self-consistency means that the design is very likely a poor one (but good self-consistency does not automatically mean its a good design). We have actually discussed this and **caveated our results** at several instances in the paper.
>
> Re. AlphaFold3's server:
> - We are restricted to **20 jobs per day**, and it is **not possible to submit batched jobs**.
> - Given these restrictions, and the fact that AF3 server came out after the NeurIPS submission deadline, would you agree that actually using AF3 for our evaluations is impossible?
>     - Here is a rough scale of how many times a structure predictor is called during our evaluation protocol:
>         - For example, take 100 test RNA backbones.
>         - Design 16 sequences for each backbone.
>         - Fold the 16 x100 designed sequences = 1600 calls per model evaluation (per one entry in the table in Appendix D)
>     - Assuming there are on average 5 authors in our team, evaluating one variant of our model would take 16 days via AF3 server. We have run experiments on ~25 models (not including further results we will be adding after rebuttals to other reviewers).
>     - Also note that there is no programmatic way to run AF3 server -- we would have been manually doing data entry.
>
> Re. 2D structure predictors: Okay, we will now be rushing to try one of these out and will report back the results honestly. **Please stay tuned.**

---

> > ### Author Response · Authors · 2024-08-12
> > **Using RNAFold instead of EternaFold for 2D self-consistency**
> >
> > > for the 2D self-consistency scores, I think it really doesn't hurt for you to give RNAfold or LinearFold a try. I doubt it would generate any significant impact on the results paper, but having these additional checks would greatly strength this paper.
> >
> > We just got results for this experiment: Replacing EternaFold with RNAFold as the 2D structure predictor lead to **unchanged results** and **did not modify the relative rankings of the models.**
> >
> > Model parameters | EternaFold scMCC | RNAFold scMCC
> > ---|---|---
> > AR, 1 state, Equivariant GNN | 0.5903 +- 0.0147 | 0.6051 +- 0.0185
> > NAR, 1 state, Equivariant GNN | 0.4337 +- 0.0324 | 0.4352 +- 0.0361
> >
> > The results are for the single-state test split, and we reported results for 3 different models trained with identical random seeds (following the same protocols as the main paper).
> >
> > We would like to add these results in the revised version of our paper, too.
> >
> > We have worked hard to alleviate your concerns as much as possible. Please let us know how to proceed.

---

### Author Rebuttal · Authors · 2024-08-06

Thank you to the reviewers for their feedback and actionable suggestions! Everyone highlighted the following positives:
- Careful data preparation and experimental evaluation (vioU, h3v6, E2ka)
- Introduction of multi-state design and representation learning (E2ka, h3v6, Hi6M)
- New design capabilities: Improved inference speed over physics-based tools (h3v6); Zero-shot mutant fitness ranking (E2ka, QQ46)
- Clear presentation and writing (vioU, E2ka, QQ46)

We have individually responded to each reviewer’s questions and concerns. We have also addressed questions common questions below.

---

## Comparison to RDesign
- Please see Appendix A, ‘Comparison to contemporaneous work’ for details re. why apples-to-apples comparison to RDesign is not possible (**no training code** and **use of random splits**, primarily). We state in detail the differences to our work in terms of methods, evaluation, and open science.
- Additionally, in Appendix D, ‘Ablation Study’, we ablate gRNAde’s architecture and compare the performance of our autoregressive model in the main paper with a **non-autoregressive variant** -- this is what RDesign uses in their architecture.
    - It is very interesting that non-autoregressive decoding improves sequence recovery but autoregressive decoding has significantly higher 2D and 3D self-consistency scores (which we care about more in real-world design scenarios). We have provided more details in the appendix.
    - We think this finding (and others from the ablation study) will be interesting to others working on biomolecule inverse folding.
- We also think these works developed concurrently but that their paper got published at a conference first. We hope this will not be held against us.

---

## On architectural novelty
- Our contributions focus on a **new application for geometric deep learning** (3D RNA design), developing careful datasets and splits, as well as rigorous experimental protocols. This aligns with broader trends across deep learning, emphasizing thorough experimentation with existing models over architecture engineering.
-  In Appendix D, we ablated the our architecture design of the encoder-decoder GVP-GNN and found **new insights** about inverse folding models. These architecture have been extremely successful for protein design (eg. ProteinMPNN uses it and has lead to a $1B startup), so we think our findings will be of broader interest.
- In the new PDF attached with the rebuttal, **we explored a new multi-state pooling function** based on [Deep Symmetric Sets](https://arxiv.org/abs/2002.08599), which is **provably more expressive than Deep Sets** when pooling over a set of features which themselves have symmetry constraints (in our case, we are pooling features from a set of RNA conformations, each of which is roto-translation and permutation equivariant). Results are shaded in blue, compared to red for Deep Sets. DSS **does not significantly improve performance** over DS on the test set, although it fits the training data better. Expressive architectures overfitting and **not generalizing to out-of-distribution data splits** has been a repeated trend across ML for structural biology, which we think further justifies our decision to keep the architecture simple.
- We think that establishing rigorous protocols and releasing **open source code** will lead to others developing better RNA inverse folding architectures in the future.

---

## On marginal improvements for multi-state gRNAde
- We agree that the results in Figure 4 show that multi-state gRNAde architectures marginally improve performance for multi-state RNAs. Just to reiterate:
    - Figure 4(a) shows overall improvements in aggregated performance across 100 test set samples, eg. sequence recovery from 0.455 → 0.484.
    - Figure 4(b) shows **why** there is improvement: at a per-nucleotide level, multi-state gRNAde has better performance for nucleotides that are locally more flexible, undergo changes in base pairing within multiple states, and are on the surface.
- Why is the performance gain only marginal? We believe there are several reasons:
    1. **Relatively fewer multi-state training samples**: There are multiple states available for 1.5K sequences out of 4K+ (see Appendix Figure 15 for visualizations). Out of these, the most interesting and highly dynamic RNAs (with larger RMSDs between states) are actually assigned to our validation and test sets during data splitting, so models are trained on less dynamic RNAs and are being **evaluated for their generalization capability** to highly dynamic RNAs.
    2. **Difficulty of the multi-state task itself**: In addition to point (1), we can further see why the multi-state split is such a difficult task by comparing the ‘Groundtruth sequence prediction baseline’ in Appendix D, Table 1 for the single-state and multi-state split.
    3. **Challenge of evaluating multi-state design**: Structural self-consistency metrics are not ideal for evaluating RNAs which undergo changes to their structure; it would perhaps be more principled (but extremely slow, expensive and intensive) to run MD simulations to validate our multi-state design models.

We would also like to share another interesting result from the appendix: **multi-state gRNAde also improves performance on the single-state test set.** See Appendix D, ‘Ablation Study’, line 600 onwards (in the table, see the columns for sequence recovery as well as 2D/3D self-consistency for the ablated variants highlighted in red). Thus, even for test set RNAs that have only one state available, training gRNAde in a multi-state manner can lead to better inverse folding performance.

Overall take away: Multi-state training seems to allow gRNAde to better understand and extract information about RNA dynamics and conformational changes, as shown by:
- Improved performance on single-state and multi-state sets.
- Source of improvement comes from nucleotides that are more structurally flexible.

---

### Author Response · Authors · 2024-08-13
**Direct comparison to RDesign checkpoint**

Dear reviewers,

We realise that 3 of you are still concerned that we have not compared directly to RDesign. (Please see our response for why we think it was justified + it is concurrent work but beat us to the race of publication.)

We have been working hard to get the RDesign codebase to work. It has been very challenging as it is impossible to reproduce the results or re-train their model, please see [this comment](https://openreview.net/forum?id=Fm4FkfGTLu&noteId=q3oYMrQ5Os) and [this comment](https://openreview.net/forum?id=Fm4FkfGTLu&noteId=6R0Gjy0EXH), or just visit their codebase and find the complete lack of documentation. After spending 6 hours, we have managed to load their checkpoint and run inference with it. Note that they also do not use any sampling (*which makes their model practically useless*), but we implemented sampling in the same way as we implemented it for our Non-autoregressive variant of gRNAde.

**Overall, RDesign significantly underperforms gRNAde on all 14 of the high-quality RNAs of interest identified by Das et al.**

**gRNAde has improved sequence recovery over Rosetta, and Rosetta has improved sequence recovery over RDesign.**

Please also note that the RDesign checkpoint almost surely has data leakage because it was trained on a random split, so we expect performance to drop further if we are able to re-train it fairly and make apples-to-apples comparison.

---

Here are the results:

Average native sequence recovery, in the same format as Figure 2(a):
- gRNAde: 0.568
- Rosetta: 0.450
- RDesign: 0.429

Full results, in the same format as Appendix Table 2:

|PDB ID|Description                               |Rosetta Recovery|RDesign Recovery|Rdesign Perplexity|RDesign 2D self-cons.|gRNAde Recovery|gRNAde Perplexity|gRNAde 2D self-cons.|
|------|------------------------------------------|----------------|----------------|------------------|---------------------|---------------|-----------------|--------------------|
|1CSL  |RRE high affinity site                    |0.44            |0.4455          |2.4220            |0.1742               |0.5720         |1.2813           |0.8645              |
|1ET4  |Vitamin B12 binding RNA aptamer           |0.44            |0.3929          |2.5460            |0.1301               |0.6250         |1.3458           |-0.0136             |
|1F27  |Biotin-binding RNA pseudoknot             |0.37            |0.3013          |2.2294            |0.2488               |0.3438         |1.6204           |0.4524              |
|1L2X  |Viral RNA pseudoknot                      |0.48            |0.3727          |2.5163            |0.3289               |0.4721         |1.3181           |0.5693              |
|1LNT  |RNA internal loop of SRP                  |0.53            |0.5556          |1.9740            |0.4169               |0.5844         |1.4337           |0.1380              |
|1Q9A  |Sarcin/ricin domain from E.coli 23S rRNA  |0.41            |0.4417          |2.3934            |0.1669               |0.5045         |1.3412           |0.0597              |
|4FE5  |Guanine riboswitch aptamer                |0.36            |0.4112          |2.6697            |0.5121               |0.5301         |1.3825           |0.9116              |
|1X9C  |All-RNA hairpin ribozyme                  |0.5             |0.3967          |2.0729            |0.3285               |0.5000         |1.3905           |0.6630              |
|1XPE  |HIV-1 B RNA dimerization initiation site  |0.4             |0.3834          |2.5047            |0.1582               |0.7037         |1.2177           |0.7768              |
|2GCS  |Pre-cleavage state of glmS ribozyme       |0.44            |0.4518          |2.4815            |0.2849               |0.5078         |1.3054           |0.4062              |
|2GDI  |Thiamine pyrophosphate-specific riboswitch|0.48            |0.3523          |2.2182            |0.3205               |0.6500         |1.2363           |-0.0252             |
|2OEU  |Junctionless hairpin ribozyme             |0.37            |0.5000          |2.4634            |0.2101               |0.9519         |1.0914           |0.7769              |
|2R8S  |Tetrahymena ribozyme P4-P6 domain         |0.53            |0.5641          |2.4050            |0.1095               |0.5689         |1.1881           |0.7281              |
|354D  |Loop E from E. coli 5S rRNA               |0.55            |0.4458          |2.3081            |0.1723               |0.4411         |1.4939           |0.0431              |

---

**Please consider revising your assessment of the paper, as we have worked hard to address all the concerns raised.**

We've tried to do honest work and release high-quality code and resources that we think will be valued by the community. We are not chasing architectural novelty; we want to build a system that is useful for a real-world problem and rigorously benchmark it.

Best regards,

Authors of gRNAde

---

### Author Response · Authors · 2024-08-13
**Summary of rebuttal discussions**

Dear Reviewers and AC,

As the discussion period will close in a few hours, we have summarized all the questions+weaknesses identified by the reviewers and how we have addressed all of them. We hope this will be useful for making your decision. Please write to us ASAP if there are any outstanding concerns.

---

**Direct comparison to RDesign public checkpoint** (Reviewers vioU, E3ka):
- We [ran inference with RDesign](https://openreview.net/forum?id=Fm4FkfGTLu&noteId=RdkCCdrrqx) and found that it **significantly underperforms** gRNAde as well as Rosetta. This is despite RDesign being trained on a random split --> test data leakage compared to gRNAde being evaluated o.o.d.
- We uncovered serious [reproducibility issues](https://openreview.net/forum?id=Fm4FkfGTLu&noteId=q3oYMrQ5Os) and [technical flaws](https://openreview.net/forum?id=Fm4FkfGTLu&noteId=6R0Gjy0EXH) with RDesign code.
- We have gone to great lengths to do this comparison to what we believe is concurrent work that beat us to publication. We hope the reviewers are happy with this.

**Apples-to-apples comparison to RDesign's architectural components**: Our ablation study already ablates key architectural choices that differ between RDesign and gRNAde; see [this comment](https://openreview.net/forum?id=Fm4FkfGTLu&noteId=OsoSako8HF) for details.

**Architectural novelty vs. applied work** (Reviewers h3v6, E2ka, Hi6M):
- We think papers not introducing new architectures are well within the scope of NeurIPS.
- We think our main technical contributions are rigorous experiments and evaluation + new insights to a novel application area. We think that it is valuable to the community to understand the capabilities of our best models on new problems.
- This is the **first work in deep learning to tackle multi-state design** and introduce multi-state representation learning for biomolecules. We also did experiments on more expressive set pooling methods (see rebuttal PDF), but found that they do not generalize any better than Deep Sets o.o.d.
- This is also the **first work to use deep learning for RNA mutant ranking given structural context**.
- We believe the gRNAde tool and codebase will unlock new capabilities for RNA designers and help machine learners build better models in the future.

---

**How the data was split** (Reviewer vioU): We used [US-align criteria](https://openreview.net/forum?id=Fm4FkfGTLu&noteId=RA6umvZYnO), which is the standard in the structure modelling/prediction community and was used in the latest CASP. There is strong precedent and this should not be counted as a weakness.

**Using different structure predictors for evaluation** (Reviewers vioU): We did [an experiment](https://openreview.net/forum?id=Fm4FkfGTLu&noteId=OmvMVAfYZC) on whether changing from EternaFold to RNAfold impacts our evaluation metrics and relative performance of models: it has no significant impact, only marginal differences within standard deviation.

**Using AlphaFold3 to evaluate or ideas from AlphaFold3 to train our models** (Reviewers vioU): AF3 came out days prior to the NeurIPS deadline and [its code or checkpoints are not available](https://openreview.net/forum?id=Fm4FkfGTLu&noteId=WPrtt5CXnl) -- we think it is not fair to ask us to use it in experiments/use ideas from it.

**Use of word 'conformer'** (Reviewer vioU): We will revise this to conformational state.

---

**Trained on small dataset** (Reviewers h3v6, QQ46): We have trained on ALL publicly available RNA 3D structures and we believe our model is useful for real-world applications. Inverse folding problem is token level, unlike structure prediction (global). We feel confident enough that gRNAde designs have been sent to experimental collaborators for testing.

**Evaluated on only 14 samples** (Reviewer h3v6): We have evaluated on 100+ RNAs across multiple different splits. This is a misunderstanding.

**Arbitrary decoding order** (Reviewer h3v6): We did an experiment in the rebuttal PDF showing that gRNAde can be trained with arbitrary decoding order with minimal performance loss.

---

**Speed claims over Rosetta** (Reviewer E2ka):
- We agree that its not possible to directly compare CPU vs GPU speed and will revise the claims. A lot of deep learning papers have made the same claim in the past -- there is strong precedent that our tools are more broadly accessible than Rosetta. Nobody can use Rosetta RNA recipes in latest builds.
- gRNAde inference on CPU is also <10s to produce designs, compared to O(hours) for Rosetta.

---

**Did we retrain baselines** (Reviewer Hi6M): This was a misunderstanding, we gave a detailed reply.

---

**Perplexity correlation** (Reviewer QQ46): We measured how well perplexity correlates with sequence and structure recovery in the rebuttal PDF.

**Ablate quantity of training data** (Reviewer QQ46): We already did this in the appendix.

**Potential negative social impact** (Reviewer QQ46): We will add this.

----

Best regards,

gRNAde Authors

---

### Decision · Program_Chairs · 2024-09-25

**Decision:**

Reject

**Comment:**

The paper introduces gRNAde,  to tackle the RNA inverse folding problem, which is a significant challenge due to RNA's potential as a therapeutic modality and its unique structural properties. The proposed method, a multi-state geometric graph neural network (GNN), constructs custom multi-graph representations of RNA and leverages an architecture consisting of a multi-state GNN encoder, a pooling layer, and an autoregressive decoder.

Strengths:
+ The authors introduce gRNAde, the first method to consider multi-state biomolecule representation.
+ The evaluation metrics are thoughtfully selected, and the experiments are conducted with fairness and rigor. The careful dataset splitting further enhances the validity of the results.
+ The paper is exceptionally well-written, with clear and thorough explanations.

Weaknesses:
- The model architecture proposed by the authors lacks innovation. The core structure of gRNAde is primarily based on a straightforward stacking of GVP-GNN layers, and its approach to handling multi-state conformations is simplistic. All components are derived from previous work, with the overall structure closely resembling that of ProteinMPNN.
- The comparison with contemporary deep learning models for RNA inverse folding is limited. The baselines used in various experiments are either outdated or overly simplistic, which may undermine the strength of the comparative analysis.
- The experimental settings could be improved. For example, the self-consistency scores and data splits could be more rigorously defined.

In their rebuttal, the authors provided additional explanations regarding the architectural novelty of their work and offered more details about the experimental settings. They also conducted new experimental comparisons with RDesign, a recently published and contemporaneous model. The reviewers appreciate the efforts made by the authors to address the concerns raised.

However, despite these efforts, Reviewer E2ka and Reviewer h3v6 remain unconvinced of the method's architectural novelty, arguing that the contribution of this paper appears to be more of an extension of existing architectures applied to RNA-related tasks. Specifically, Reviewer E2ka believes that it would be more appropriate to retrain the model architecture by RDesign on the dataset used in this paper, rather than directly testing the model using a pre-existing checkpoint.

Given that RDesign was published at ICLR 2024 and was accessible on OpenReview during the review period, it would be reasonable to explicitly present the comparison with it in both the Introduction and Methodology sections, rather than relegating it to the appendix. Since addressing the discussion and experimental comparison with RDesign would require substantial modifications, this paper is not ready for publication in its current form. The authors are strongly encouraged to consider the reviewers' feedback and resubmit to the next conference.